# Regulation and Molecular Mechanism of *TLR5* on Resistance to *Escherichia coli* F18 in Weaned Piglets

**DOI:** 10.3390/ani9100735

**Published:** 2019-09-27

**Authors:** Chaohui Dai, Li Yang, Jian Jin, Haifei Wang, Shenglong Wu, Wenbin Bao

**Affiliations:** 1Key Laboratory for Animal Genetics, Breeding, Reproduction and Molecular Design, College of Animal Science and Technology, Yangzhou University, Yangzhou 225009, China; dx120170081@yzu.edu.cn (C.D.); 18705272191@163.com (L.Y.); jianj1127@163.com (J.J.); hyfiwang@yzu.edu.cn (H.W.); slwu@yzu.edu.cn (S.W.); 2Joint International Research Laboratory of Agriculture & Agri-Product Safety, Yangzhou University, Yangzhou 225009, China

**Keywords:** pig, diarrhea, *TLR5* gene, RNA-seq, DNA methylation, *E. coli* resistance

## Abstract

**Simple Summary:**

Piglet diarrhea is the most common type of disease on many large-scale pig farms, where it causes serious economic losses. Abuses of vaccines and veterinary medicine are very common, which not only lead to the emergence and prevalence of drug-resistant bacteria, but also to a decline in the quality of livestock products and economic benefits. Therefore, improving the resistance of piglets to diarrhea by genetic means is a common goal of modern breeding scientists. This study aims to reveal the molecular mechanism of regulating piglet diarrhea via the important candidate gene *TLR5* from the perspective of epigenetic factors.

**Abstract:**

Toll-like receptor 5 (TLR5) plays an important role in immune system. In this study, we performed transcriptome analysis of the duodenum in *E. coli* F18-resistant and -sensitive Sutai weaned piglets and analyzed the differential expression of *TLR5*. The cellular localization of TLR5 was investigated, and the effect of *TLR5* expression on *E. coli* invasion was evaluated after pig small intestinal epithelial cell lines (IPEC-J2) were stimulated by *E. coli*. The results showed that TLR5 expression level in duodenum and jejunum were significantly higher in *E. coli* F18-sensitive than in *E. coli* F18-resistant piglets. TLR5 protein was mainly expressed in the cytoplasm and cell membrane. The expression of genes associated with the TLR5 signaling pathway were significantly higher in *TLR5*-overexpressed cells than in control cells. Bacterial adhesion was higher in *TLR5*-overexpressed cells than in blank cells and lower in *TLR5* interference than in blank cells. The core promoter region of *TLR5* included two CpG islands and 16 acting elements. The methylation of the mC-6 site in the second CpG island of the promoter region had a regulatory effect on *TLR5* expression. Therefore, *TLR5* plays an important regulatory role on *E. coli* invasion. Low expression of *TLR5* inhibited the immune response and decreased cell damage, which was conducive to the resistance to *E. coli* stimulation. In conclusion, this study preliminarily revealed the molecular mechanism of *TLR5* gene regulating the resistance of piglets to *Escherichia coli*, and provided a new candidate gene for screening *Escherichia coli* resistance markers in pigs.

## 1. Introduction

Toll-like receptors (TLRs) are involved in the innate immune system. TLRs recognize pathogen-associated molecular patterns (PAMPs) through pattern recognition receptors leading to the production of pro-inflammatory factors. Therefore, TLRs play an important role in host immune responses and inflammatory processes [1,2]. TLR5, a member of the TLR family, recognizes flagellin, which is the main component of flagella in Gram-negative bacteria. The binding of TLR5 to flagellin results in the activation of the nuclear transcription factor beta (NF-κB) and stimulates the production of TNF-α, IL-1β, IL-8, and other inflammatory cytokines [3,4,5]. Flagellin may be the only bacterial PAMP that specifically activates TLR5 and stimulates the production of proinflammatory cytokines such as IL-1β and IL-8 by activating the NF-κB signaling pathway downstream [6]. Studies have shown that TLR5 is mainly involved in MyD88-mediated Toll-like signaling pathway, inflammatory bowel disease, pathogenic *Escherichia coli* infections, salmonellosis, and legionellosis [7,8,9,10,11].

The porcine *TLR5* gene has a Leucine-Rich Repeats (LRR) domain and TIR domain (toll/interleukin-1 receptor domain), which are characteristic of TLRs [12]. In addition, porcine TLR5 protein had a high homology with other mammalian TLR5 proteins and was ubiquitously expressed in the kidney, liver, lung, small intestine, spleen, thymus, and other organs [13]. What is more, the expressed porcine TLR5 protein generates an adequate immune response [14]. Studies have shown that the flagellin-TLR5 complex stimulates small intestinal epithelial cells to secrete IL-6 and IL-8 and other proinflammatory cytokines [15]. In addition, porcine *TLR5* was reported to play an important regulating role in *Salmonella choleraesuis*, swine erysipelas and diabetic nephropathy [16,17,18]. Yang et al. [19] showed that SNPs on *TLR5* were correlated with cytokine abundance in vivo and that the expression of *TLR5* was significantly correlated with IL-10 levels. Therefore, porcine *TLR5* plays an important regulatory role in the innate immune response against exogenous antigens. Dominguez et al. [20] identified one indel variant in the predicted *TLR5* promoter sequence, which contributed to an additional STAT binding site and an increase in promoter activity. Compared with Western breeds, Chinese domestic pig breeds have higher genetic diversity and more haplotypes [21]. In this study, *TLR5* and the Toll-like signaling pathway were determined by transcriptome analysis of *E. coli* F18-resistant and -sensitive weaned Sutai piglets (Meishan × Duroc). In view of the important biological functions of *TLR5*, we studied the relationship between *TLR5* and *E. coli* resistance of weaned piglets as well as epigenetic modification in this study. We aimed to understand the molecular mechanism of *TLR*5 gene regulation on *E. coli* resistance in weaned piglets.

## 2. Materials and Methods

### 2.1. Ethics Statement

Animal experiments were approved by the Institutional Animal Care and Use Committee of the Yangzhou University Animal Experiments Ethics Committee (permit number: SYXK [Su] IACUC 2012-0029). All experimental procedures were performed in accordance with the Regulations for the Administration of Affairs Concerning Experimental Animals approved by the State Council of the People’s Republic of China. Pigs were euthanized by pentobarbital sodium injection (dose of 100 mg/kg).

### 2.2. Animals

The experimental pigs used in this study included *E. coli* F18-resistant and -sensitive Sutai pigs from the Sutai Pig Breeding Center in Suzhou (Jiangsu Province, China). We screened three resistant and three sensitive animals of similar birth weight, weaning weight, body shape, and coat color using a V-type secretion system [22]. All pigs were housed in the same environment. At the age of 35 days, when piglets are most susceptible to *E. coli* F18, the animals were sacrificed, and intestinal tissues (duodenum and jejunum) were excised and transferred to liquid nitrogen.

### 2.3. Primer Design and Sequence Synthesis

Quantitative PCR (qPCR) primers for *TLR5* based on its coding sequence, PCR primers for the promoter region of *TLR5* and Bisulfite Sequencing PCR (BSP) primers for methylation detection were designed. All primers (Appendix A) were synthesized by Sangon Biotech Co., Ltd. (Shanghai, China).

### 2.4. Transcriptome Analysis of Duodenum in E. coli F18-Resistant and Sensitive Piglets

Total RNA was extracted from the duodenum of three resistant individuals (SR1, SR2, and SR3) and three sensitive individuals (SS1, SS2, and SS3) using Trizol. After assessing RNA quality, RNA library was established according to standard procedure [23]. Then, we used Hiseq 4000 for sequencing analysis, and the amount of data was about 6G clean data for each sample. We performed quality control of the raw reads obtained from Hiseq4000 sequencing. NGS QC Toolkit v2.3.3 software [24] was used to remove low-quality fragments. Bowtie2 software [25] was used to create a reference genome index. The Tophat2 (http://tophat.cbcb.umd.edu/) software (University of Maryland, Baltimore, MD, USA) was used to compare the filtered reads to the reference genome. Gene annotations (including structural and functional annotations) for the predicted mRNA sequence were conducted, and the differentially expressed genes were identified. Based on the differentially expressed genes, cluster analysis, GO function significance enrichment analysis, and pathway significance enrichment analysis were performed.

### 2.5. Total RNA Extraction, cDNA Synthesis, and qPCR

We extracted total RNA from the duodenum and jejunum of *E. coli* F18-resistant and -sensitive Sutai piglets and IPEC-J2 cells using Trizol. The purity and concentration of total RNA were assessed using 1% formaldehyde agarose gel electrophoresis and NanoDrop 1000 (General Electric Company, USA). RNA was stored at −70 °C. cDNA was synthesized according to kit of HiScript II Q RT SuperMix for qPCR (Vazyme Biotech Co., Ltd., Nanjing, China). The resulting cDNA was stored at −70 °C. QPCR was conducted by kit of AceQ qPCR SYBR Green Master Mix (Vazyme Biotech Co., Ltd., Nanjing, China). Relative expression was calculated by the 2^−ΔΔCt^ method [26] and interference efficiency was calculated by 1−2^−ΔΔCt^.

### 2.6. Western Blot Analysis

Total protein from the duodenum and jejunum of *E. coli* F18-resistant and -sensitive Sutai piglets and IPEC-J2 cells was extracted using RIPA Lysis Buffer (Cwbio, Ltd., Taizhou, China) and analyzed using the BCA kit (Thermo Fisher Scientific Inc, Massachusetts, USA). SDS-PAGE (polyacrylamide electrophoresis) of the protein samples was performed at 120 V for 90 min in a 10% gel. Protein sample was transferred to a PVDF membrane (polypropylene difluoroethylene membrane) and immunoblotted with the relevant antibody. Blocking solution and anti-TLR5 (1:800) and β-actin (1:4000) antibodies were added. HRP (horseradish peroxidase labeled antibody, 1:5000) was used as a secondary antibody, and β-actin protein was used as a reference.

### 2.7. Cell Localization of TLR5 Protein

The localization of TLR5 was analyzed by indirect immunofluorescence. Cell slides were removed from an incubator and gently washed three times with PBS. Subsequently, the cells were fixed with 4% paraformaldehyde at 4 °C for 15 min and gently washed three times with PBS. Triton X-100 was added to the cells, which were subsequently washed three times with PBS. Following the removal of PBS, we added goat blocking serum and incubated the cells for 1 h at room temperature. TLR5 antibody (1:500) was added to the cells, which were subsequently incubated at 37 °C for 2 h and stored at 4 °C overnight. The cell slides were gently washed with PBS three times. Secondary antibody IgG-HRP (1:3000) labeled by red fluorescent was added to the cells and incubated at room temperature for 1 h. All subsequent procedures were performed minimizing exposure to light. When antibody was removed, the cells were washed three times with PBST. DAPI was added for staining (1:1000), and cells were incubated for 5 min and washed lightly with PBS three times. The cell slides were fixed with anti-fade fluorescence mounting medium and immediately observed under a fluorescence microscope (Olympus Fluorescence Inverted Microscope IX73, Olympus Corporation, Tokyo, Japan).

### 2.8. Establishment of Stable Cell Lines with TLR5 Interference and Overexpression

Pig small intestinal epithelial cell lines (IPEC-J2) were seeded onto 6-well plates at 1 × 10^6^/well and cultured at 37 °C and 5% CO_2_ with DMEM/F12 medium supplemented with 10% fetal bovine serum (FBS). When cells reached 80% density, lentivirus of interference groups (PGLV3-TLR5-1, PGLV3-TLR5-2, pGLV3-TLR5-3, pGLV3-TLR5-4, and negative control pGLV3-TLR5-NC) and overexpression groups (pGLV5-TLR5 and negative control pGLV5-NC) were added to the cells. The normal cell group (blank group) without transfection of lentivirus was prepared at the same time. Each treated group had three replicates. Cells were incubated overnight and analyzed for fluorescence the following day.

Following a 48-h transfection period, total RNA and total protein of different treatment groups were extracted using Trizol. PCR was performed to evaluate the expression of target genes in cells. Additionally, TLR5 protein expression was analyzed by western blot.

According to the mRNA and protein expression of *TLR5*, the lentivirus with the highest interference efficiency was selected. The over-expression efficiency of *TLR5* was verified at the same time. After continuous culture of cells, we selected positive monoclonal cells using blasticidin until the expression of fluorescence was approximately 100% and the expression level of *TLR5* was stable. Finally, IPEC-J2 cells with *TLR*5 silencing and *TLR5* overexpression were obtained for further functional verification.

### 2.9. Stimulation of Cells with E. coli

IPEC-J2 cells treated with *TLR5* interference group, *TLR5* overexpression group, and blank control group were seeded onto 12-well plates at 5.0 × 10^5^ cells/well and incubated for 24 h at 37 °C and 5% CO_2_. Each treatment sample had three repeats. *E. coli* F18ab, *E. coli* F18ac, and *E. coli* K88ac fimbriae standard strains were inoculated to LB culture medium and incubated for 12 h at 200 r/min on a rocking platform. The bacteria were diluted to 1.0 × 10^9^ CFU/mL with cell culture medium. Culture medium (1.0 mL) was added to 12-well culture plates and incubated for 3 h. The effect of *TLR5* overexpression and interference on cells with *E. coli* invasion was detected by ELISA, Gram-stain, scanning electron microscopy (SEM) and indirect immunofluorescence.

The expression levels of *TLR5*, *MyD88*, and *TNF-α* in IPEC-J2 cells were detected by qPCR and Western blot. Proinflammatory cytokines (TNF-α, IL-6, IL-8, IL-10, and IL-12) in the cell culture supernatant were detected by ELISA.

Cells stimulated by *E. coli* were slightly washed three times with PBS to remove non-adhered bacteria. Cells were stained with crystal violet (early staining) for 1 min and washed with water. Subsequently, the cells were stained with iodine solution (mordant) for 1 min. Alcohol (95%) was used to decolorize the solution. Finally, cells were stained with carbonic acid reddish dilution for 30 s. After washing and drying, the bacteria adhered to the cells were observed under an oil immersion microscope (×400).

Cells stimulated by *E. coli* were slightly washed three times with PBS to remove non-adhered bacteria. Glutaraldehyde solution (1 mL, 4%) was added to each well, and the cells were incubated for 1 h at 4 °C. The immobilized samples were washed three times with PBS. The samples were dehydrated with different alcohol concentrations for 10 min each. After dehydration, the samples were immersed in tert-butyl alcohol, and the mixture was incubated three times at 28 °C for 15 min each time. Following the last incubation with tert-butyl alcohol, the cells were stored at 4 °C for 25 min and transferred to a vacuum drier for 30 min. Silicon and sample pallets were pasted using a conductive double-sided tape and analyzed by scanning electron microscopy after spraying with gold.

Cells stimulated by *E. coli* were slightly washed three times with PBS to remove non-adhered bacteria. The cells were fixed with 4% paraformaldehyde at 4 °C for 15 min and slightly washed three times with PBS. Triton X-100 was used to treat cells for 10 min, and PBS was used to wash the cells three times. When PBS was removed, goat blocking serum was added to the cells and incubated for 1 h. *E. coli* antibody (1:500) was added to cells, incubated at 37 °C for 2 h, and placed at 4 °C overnight. Subsequently, the cell slides were washed three times with PBS. Secondary antibody IgG-HRP (1:3000) labeled with red fluorescent was added to the cells, which were incubated at room temperature for 1 h. Subsequent procedures were performed minimizing exposure to light. When the antibody was removed, cells were washed three times with PBST. Following the addition of DAPI (1:1,000), the cells were incubated for 5 min and washed three times with PBS. The cell slides were fixed using anti-fade fluorescence mounting medium and immediately analyzed by confocal microscopy (Operetta CLS, PerkinElmer, USA).

### 2.10. Determination of TLR5 Promoter Region and Analysis of Acting Elements

The 2000-bp sequence upstream of the transcription start site of the *TLR5* genome sequence was used as the template. Primers were designed (Appendix A) to amplify the different sections of promoter fragments truncated at 3′ end of the sequence. The PCR products of seven different primers were confirmed and purified. Purified PCR products and pCpGL-basic vector were digested using *SpeI* and *NcoI.* Digestion products were purified and connected using T4 ligase at 16 °C overnight. The ligation products were transformed into competent cells DH5a and incubated in antibiotic-free LB medium with low salt for 1 h at 220 rpm. The bacterial solution was transferred onto a plate containing zeocin and incubated overnight at 37 °C. Monoclonal bacteria was selected and incubated in LB medium with zeocin for 5 h. PCR was performed using the bacterial solution. The PCR products were detected by agarose gel electrophoresis and sequenced when the electrophoresis bands were correct. The sequencing vector primer 5′–3′ was CTCTTTGTTCAGCTCTCTGTTT. The confirmed recombinant plasmids were purified and amplified for further analysis. These recombinant vectors were labeled as –2000 bp–pCpGL, –1500 bp–pCpGL, –1000 bp–pCpGL, –750 bp–pCpGL, –500 bp–pCpGL, –250 bp–pCpGL, and –100 bp–pCpGL.

IPEC-J2 cells were seeded onto 12-well plates at 1 × 10^5^/well and incubated at 37 °C and 5% CO_2_ in the presence of DMEM/F12 medium supplemented with 10% FBS. When cells reached 80% density, seven recombinant vectors were co-transfected with pRL-TK vector into cells. After transfection for 48 h, dual luciferase activity assay was performed. For sections with significant differences in dual luciferase activity, the core promoter region of *TLR5* was predicted using BDGP software (http://www.fruitfly.org/seq_tools/promoter.html), and the transcription factor binding domain of the *TLR5* gene promoter was analyzed using Alibaba 2 software (http://gene-regulation.com/pub/programs/alibaba2/index.html).

### 2.11. Detection of Methylation of CpG Island in the TLR5 Promoter Region and Its Relationship with mRNA Expression

CpG islands of the *TLR5* promoter region were predicted using the MethPrimer software (http://www.urogene.org/cgi-bin/methprimer/methprimer.cgi). The total predicted fragment at 1000 bp and predicted parameter were set as follows, the length of CpG island > 100 bp, GC content > 50%, and CpGo/e > 0.6. To detect the methylation level in the *TLR5* promoter region and its correlation with mRNA expression, DNA samples of duodenum and jejunum tissue were used as experimental material. Genomic DNA was transformed using the EpiTect Fast DNA Bisulfite kit (QIAGEN, Hilden Germany), and the PCR primers were designed based on the sequence of the sulfite transformation using the MethPrimer software. The PyroMark PCR kit (QIAGEN, Hilden, Germany) was used to amplify the transformed DNA. The target fragment was purified using the universal DNA purification extraction kit (Tiangen Biotech Co. Ltd., Beijing, China). Purified products were ligated with T vector to transform competent DH5a cells and incubated in antibiotic-free LB medium with low salt for 1 h at 220 rpm. The bacterial solution was transferred onto a plate containing zeocin and incubated overnight at 37 °C. Monoclonal bacteria were selected and incubated in LB medium with zeocin for 5 h. Ten positive clones of each plate were selected. The sequence of the methylated DNA was compared with the original DNA sequence to find CG sites that were methylated. The correlation between methylation degree and mRNA expression was analyzed using GraphPad Prism 6.

### 2.12. Statistical Analysis

Transcript gene levels were analyzed by the 2^−ΔΔCt^ method [25] and normalized to *GAPDH* levels. Differences in gene expression and cytokine levels were analyzed by ANOVA using SPSS 16.0 software, and LSD was used for post-hoc test. All experimental sample had three replicates, and the results were presented as mean ± standard deviation.

## 3. Results

### 3.1. Transcriptome Analysis of Duodenum of E. coli F18-Resistant and Sensitive Sutai Piglets

To screen differentially expressed genes related to resistance to *E. coli*, we used RNA-seq to compare transcripts of duodenum between *E. coli* F18-resistant and -sensitive Sutai piglets. There were 238 differentially expressed genes (DEGs) including 112 up-regulated DEGs and 126 down-regulated DEGs (Appendix A, some DEGs are shown in Table 1). The results indicated that the DEGs had two GO classifications including biological processes and molecular functions, which involved the “immune system process” and “immune response” (Appendix A). The screened DEGs were involved in 20 pathways including immune-related pathways such as Toll-like receptor signaling pathway (including *TLR5* and *IL1β* gene; Appendix A).

### 3.2. Differential Expression of TLR5 in Intestinal Tissues

To detect the differential expression of TLR5 in intestinal tissues, The mRNA expression levels of *TLR5* of intestinal tissues in *E. coli* F18-resistant and -sensitive Sutai piglets were analyzed by qPCR. The results showed that *TLR5* expression levels in the duodenum was significantly higher in sensitive individuals than in resistant individuals (*p* < 0.05), *TLR5* expression levels in the jejunum was extremely significantly higher in sensitive individuals than in resistant individuals (*p* < 0.01). Additionally, western blot results showed that TLR5 expression was higher in sensitive individuals than in resistant individuals (Figure 1).

### 3.3. Localization of TLR5 Protein on IPEC-J2 Cells and Establishment of Cell Lines with TLR5 Interference and Overexpression

To identify the molecular function of TLR5, the localization of TLR5 in IPEC-J2 cells was analyzed by indirect immunofluorescence. The results showed that TLR5 was mainly expressed in the cytoplasm and cell membrane (Figure 2A). The transfected lentivirus of *TLR5* overexpression and interference in IPEC-J2 cells expressed green fluorescent protein (Figure 2B), indicating that the lentivirus was efficiently integrated into IPEC-J2 Cells. Total RNA and protein were collected after 48 h to analyze the interference and overexpression efficiencies of different lentiviral vectors. The results showed that LV5-TLR5 overexpressed the transcription level of *TLR5* (1700×; Figure 2C). The interference efficiencies of LV3-TLR5-1, LV3-TLR5-2, LV3-TLR5-3, and LV3-TLR5-4 in IPEC-J2 cells were 22.80%, 30.13%, 73.85%, and 45.09%, respectively (Figure 2D). Protein expression levels were consistent with transcription levels (Figure 2E). The IPEC-J2 cells with the highest efficiency of pGLV3-TLR5-3 and LV5-TLR5 were cultured to establish the IPEC-J2 cell line with *TLR5* silencing and overexpression.

### 3.4. Expression Changes of MyD88 and TNF-α in TLR5 Signaling Pathway and the Release of Downstream Inflammatory Cytokines after E. coli Stimulation

To study the effect of *E. coli* stimulation on the TLR5 signaling pathway, the expression of *MyD88* and *TNF-α* in the TLR5 pathway were analyzed by qPCR and Western blot. The results (Figure 3) showed that the expression of three genes (*MyD88*, *TNF-α*, and *TLR5*) were significantly higher in TLR5-overexpressed cells than in control cells, while the expression level of the three genes in TLR5 interference cells were significantly lower than that in control cells (*p* < 0.05 or *p* < 0.01). The levels of proinflammatory cytokines TNF-α, IL-6, IL-8, IL-10, and IL-12 in the cell culture supernatant were measured by ELISA (standard curve shown in Appendix A). The results (Figure 4) showed that levels of cytokines were significantly higher in TLR5-overexpressed cells than in interference cells, and the levels of cytokines in the interference cells after stimulated by *E. coli* F18ab, *E. coli* F18ac, and *E. coli* K88ac tended to the levels of control cells without bacterial stimulation (*p* < 0.05 or *p* < 0.01).

### 3.5. Effect of TLR5 Overexpression and Interference on Cells with E. Coli Stimulation

The effects of TLR5 overexpression and interference on the adhesion of *E. coli* to IPEC-J2 cells were investigated by Gram staining, scanning electron microscopy (SEM), and indirect immunofluorescence. Gram staining results showed that the number of bacterial adhesion was greater in *TLR5*-overexpressed cells than in blank cells. Some cells had considerable damage, while the number of bacterial adhesion was lower in *TLR5* interference than in blank cells (Figure 5A). Based on SEM, bacterial adhesion was higher in *TLR5*-overexpressed cells than in blank cells. Some cells had some damage. The number of bacterial adhesion was lower in *TLR5* interference than in blank cells (Figure 5B). Indirect immunofluorescence results showed that the expression of bacteria was higher in *TLR5*-overexpressed cells than in blank cells, while the expression of bacteria in *TLR5* interference was lower than that of blank cells (Figure 5C).

### 3.6. Molecular Mechanisms of TLR5 Expression Regulation

In this study, we first predicted the core promoter region and CpG island of *TLR5* using the double luciferase activity assay and bioinformatic software (the amplification results of promoter methylation are presented in Appendix A). We predicted three core promoter regions of *TLR5*, and they were located at 13–62 bp, 255–304 bp, and 864−913 bp upstream of the transcription start site, including two CpG islands and sixteen acting elements (Figure 6). Results of the methylation CpG island in the promoter region showed that there were six CG sites in CpG1 and three methylation sites with no significant correlation between methylation and mRNA expression at each locus (Figure 7A,B). There were fifteen CG sites in CpG2, and methylation was different in each site. Methylation of the mC-6 CG site was significantly correlated with mRNA expression (r = −0.95 and *p* = 0.04; Figure 7C,D).

## 4. Discussion

*E. coli* F18 can combine with brush border receptors on the small intestinal epithelium and release enterotoxins, which could result in diarrhea in weaned piglets. Therefore, the pathogenicity of *E. coli* F18 depends on whether the small intestine epithelial cells of piglets express the corresponding receptors [27]. Coddens et al. [28] reported that expression levels of *E. coli* F18 receptor increased in Landrace with age, especially from 0 to 3 weeks, and remained stable from 3 to 23 weeks of age. Weaned piglets (5 to 6 weeks of age) are not sensitive to hemolytic *E. coli,* because the receptor is not expressed and antibodies in breast milk confer protection [29]. Therefore, we selected 35-d old weaned piglets with phenotypic and autoimmune characteristics that were sensitive to *E. coli* F18 stimulation. In this study, we selected *E. coli* F18-resistant and -sensitive Sutai weaned piglets for transcriptome analysis and identified *TLR5* gene. TLR5 belongs to the TLRs signal pathway. The TLRs family recognize conserved microbial structures (such as bacterial lipopolysaccharides) and activates signaling pathways that result in immune responses caused by microbial infections [30]. TLRs detect microbial populations in the gut and initiate proinflammatory signaling pathways against microbial pathogens [31]. This study performed transcriptome analysis in duodenum of *E. coli* F18-resistant and -sensitive Sutai weaned piglets and screened *TLR5* gene and the Toll-like signaling pathway. In view of the important biological functions of *TLR5*, this study investigated the relationship between *TLR5* and *E. coli* resistance of weaned piglets to understand the molecular mechanism of *TLR*5 regulation on *E. coli* resistance in weaned piglets.

We detected the expression levels of *TLR5* in small intestine tissues (duodenum and jejunum) of *E. coli* F18 -resistant and -sensitive Sutai weaned piglets. The results showed that the expression of *TLR5* in duodenum and jejunum were lower in resistant than in sensitive individuals, which revealed that the expression of *TLR5* was related to *E. coli* resistance and that its low expression might be beneficial to weaned piglets against *E. coli* stimulation. To further verify the relationship between the expression of *TLR5* and *E. coli* invasion, we established an IPEC-J2 cell line with *TLR5* interference and overexpression. The results showed that the overexpression of *TLR5* increased *E. coli* adhesion, cell damage, and cytokine secretion. Moreover, the interference of *TLR5* reduced *E. coli* adhesion and cell damage to a certain extent. These results were consistent with the biological function of TLR5 as a bacterial receptor [32]. As a member of the Toll-like receptor family, TLR5 plays an important role not only in the recognition of flagellin, but also in the regulation of resistance to *E. coli* in weaned piglets. As for the mode of *TLR5* gene expression regulating the resistance to *E. coli*, it is necessary to further analyze the molecular mechanism of *TLR5* gene regulation.

There are many ways to regulate gene expression, including transcription levels, post-transcription levels, and translation levels [33,34,35]. Among them, the methylation of the DNA promoter region is an important means for regulating genomic function [36,37]. Therefore, we conducted a series of studies on the *TLR5* gene promoter. In this study, the core promoter region and CpG island of *TLR5* gene were identified and the methylation level of the mC-6 site in the second CpG island of the promoter region had a negative regulatory effect on the expression of *TLR5*. By bioinformatics prediction, the mC-6 CG site was located in the binding domain of transcription factor Sp1. So we speculated that the methylation of the mC-6 CG site in the second CpG island of the *TLR5* promoter inhibited the binding of the transcription factor Sp1, which inhibited the expression of *TLR5* and ultimately affected the resistance of *E. coli*. However, it remained to be further verified in the future.

## 5. Conclusions

We identified an important gene *TLR5* related to the resistance to *E. coli* in Sutai weaned piglets. The low expression of *TLR5* gene might be beneficial to the resistance of piglets to *E. coli*. We speculated that the methylation of the mC-6 CG site in the second CpG island of the *TLR5* gene promoter inhibited the binding of the transcription factor Sp1, which inhibited the expression of the *TLR5* gene and ultimately affected the resistance of *E. coli*. In conclusion, this study preliminarily revealed the molecular mechanism of *TLR5* gene regulating the resistance of *Escherichia coli*, and provided a new candidate gene for screening *Escherichia coli* resistance markers in pigs.

## Figures and Tables

**Figure 1 animals-09-00735-f001:**
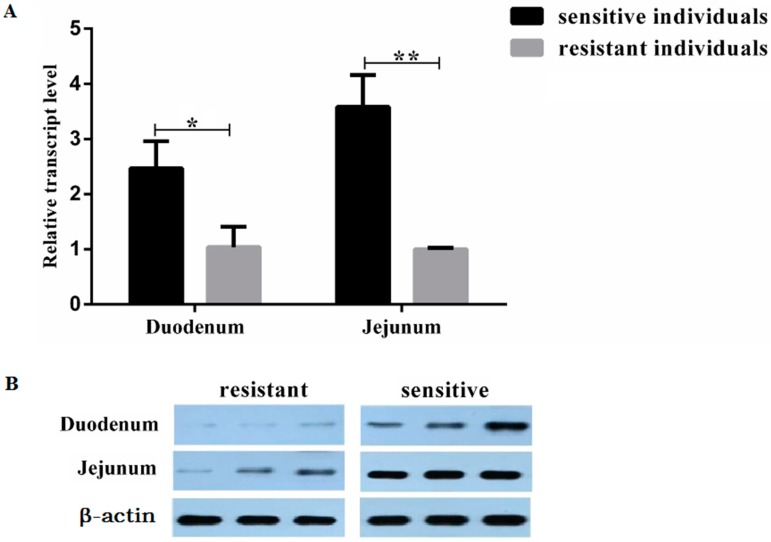
*TLR5* expression in intestinal tissues of *E. coli* F18-resistant and -sensitive Sutai piglets. Toll-like receptor 5 (TLR5) expression was assessed by quantitative PCR (qPCR) (**A**) and western blot (**B**). The experiments were divided two groups (resistant individuals and sensitive individuals) and three replicates for each group, the results of qPCR were showed as “average ± standard deviation”. *, *p* < 0.05; **, *p* < 0.01.

**Figure 2 animals-09-00735-f002:**
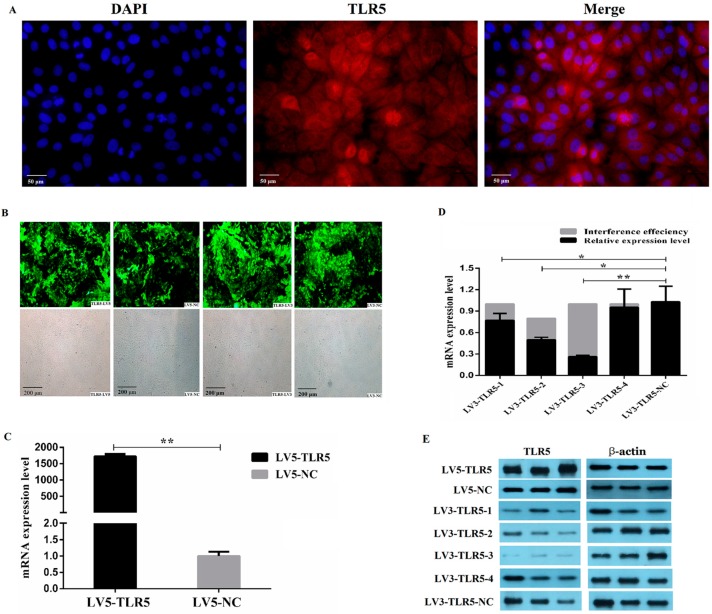
Cellular localization of *TLR5* protein in intestinal epithelial cell lines (IPEC-J2) cells and establishment of IPEC-J2 cell lines with TLR5-RNA interference (RNAi) and overexpression. (**A**) Cellular localization of *TLR5* protein in IPEC-J2 cells by indirect immunofluorescence. Blue fluorescence represents nucleus stained by DAPI; red fluorescence represents cytoplasm and cell membrane combined with *TLR5* antibody. Cells were observed by fluorescence microscopy (200×). (**B**) Expression of green fluorescent protein (GFP) in IPEC-J2 after cells were infected by lentivirus solution for 24 h (100×). (**C**) *TLR5* mRNA levels in LV5-TLR5 and LV5-NC measured by qPCR. **, *p* < 0.01. (**D**) *TLR5* mRNA levels in LV3-TLR5-1, LV3-TLR5-2, LV3-TLR5-3, LV3-TLR5-4, and LV3-TLR5-NC measured by qPCR. *, *p* < 0.05; **, *p* < 0.01. (**E**) Protein expression levels of *TLR5* in cells subjected to seven lentivirus types determined by Western blot.

**Figure 3 animals-09-00735-f003:**
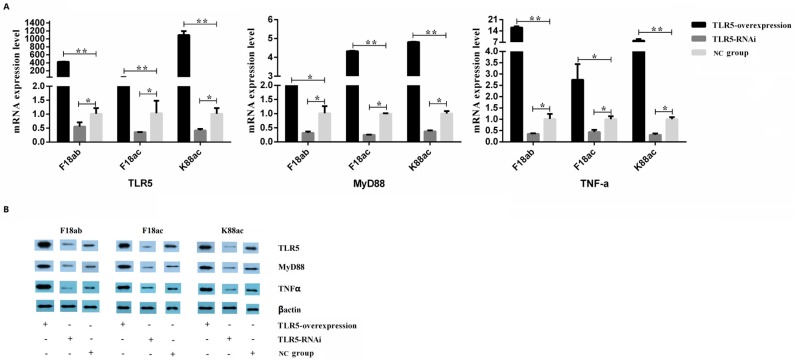
Expression levels of *TLR5*, *MyD88*, and *TNF-α* in IPEC-J2 cells following *E. coli* stimulation. Expression differences of *TLR5*, *MyD88*, and *TNF-α* were detected by qPCR (**A**) and western blot (**B**). *, *p* < 0.05; **, *p* < 0.01.

**Figure 4 animals-09-00735-f004:**
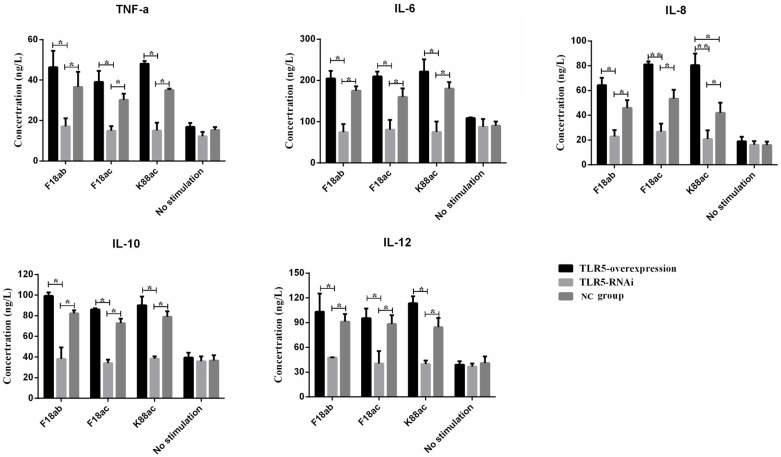
Release level of inflammatory cytokines from IPEC-J2 cells following *E. coli* stimulation. Inflammatory cytokines including *TNF-α*, IL-6, IL-8, IL-10, and IL-12 was detected by ELISA. *, *p* < 0.05; **, *p* < 0.01.

**Figure 5 animals-09-00735-f005:**
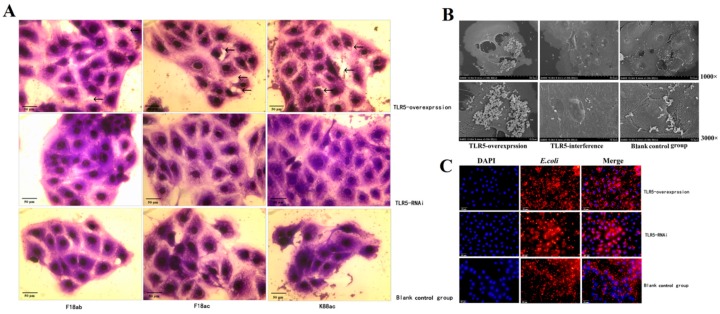
Evaluation of *E. coli* adhesion to IPEC-J2 cells. (**A**) Results of Gram staining. Cells were observed under an optical microscope (400×). Arrows represent where the cells were damaged. (**B**) Results of scanning electron microscope (SEM). Cells were observed under a scanning electron microscope (1000× and 3000×). (**C**) Results of indirect immunofluorescence. Blue fluorescence represents nucleus stained by DAPI; red fluorescence represents cytoplasm and cell membrane combined with *E. coli* antibody. Cells were observed under a fluorescence microscope (100×).

**Figure 6 animals-09-00735-f006:**
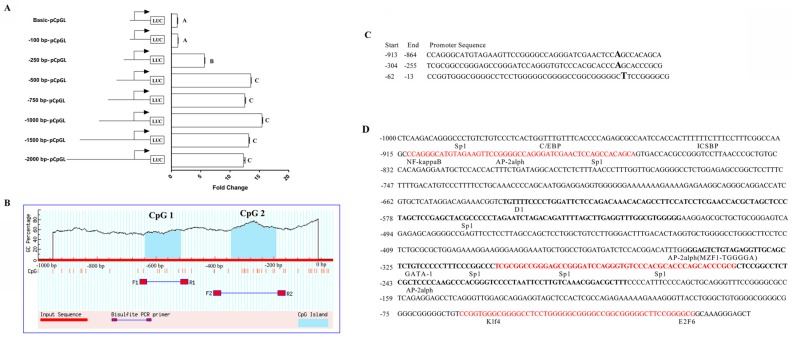
Activity verification of *TLR5* promoter region and prediction of CpG island and acting elements. (**A**) Detection of dual luciferase activity in different regions of the *TLR5* promoter region. Luciferase activity was calculated as the ratio between firefly luciferase activity (Rn) and perineurin luciferase activity (Ff). The abscissa represents fold-change in the luciferase activity of the promoter region compared to the control plasmid (Basic-pCpGL). The ordinate represents the plasmid corresponding to the different truncated segments. Different capital letters indicate significant differences (*p* < 0.01). (**B**) CpG island prediction at 1000 bp upstream of the transcription start site. (**C**) Prediction of the core promoter region based on BDGP software. (**D**) Analysis of acting elements in the promoter region. The red letters represent the core promoter region, bold letters represent CpG island sequences, and underlined letters represent the binding sites of acting elements.

**Figure 7 animals-09-00735-f007:**
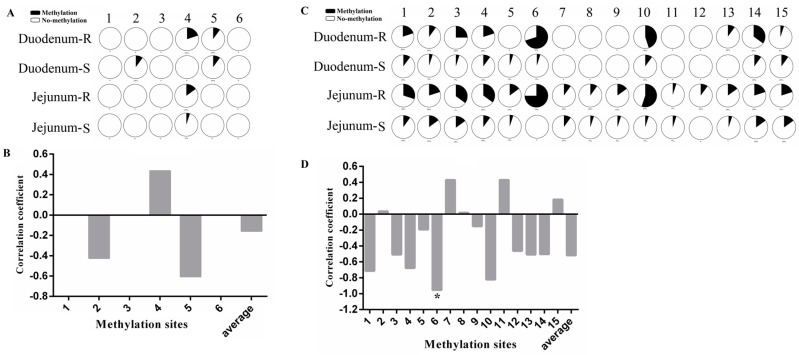
Methylation analysis of CpG islands in the *TLR5* promoter region and correlation with mRNA expression. (**A**,**B**) represent methylation analysis for single site of CpG1 and its correlation analysis with mRNA expression. (**C**,**D**) represent methylation analysis for single site of CpG2 and its correlation analysis with mRNA expression. * indicates significant difference (*p* < 0.05).

**Table 1 animals-09-00735-t001:** Partial list of differentially expressed genes in Sutai weaning piglets resistant and sensitive to *E. coli* F18 *.

Transcript	Gene Name	SR_FPKM	SS_FPKM	log2 (fold change)	*p*-Value
NM_001123202.1	*TLR5*	4.7384	7.83192	−0.72497	0.01525
NM_214055.1	*IL1β*	0.567332	1.3358	−1.23544	0.0291
NM_001206441.1	*TAP2*	1.84134	3.91261	−1.08738	0.0028
NM_001123127.1	*HSP70*	51.9633	24.4701	1.08647	0.00955
XM_005666775.1	*CXCL11*	2.72821	5.54031	−1.02201	0.0195
NM_001038004.1	*MMP9*	4.41083	2.18953	1.01043	0.0062
NM_001160080.1	*DGAT2*	38.0486	19.7141	0.948614	0.0179
NM_001244717.1	*SLC13A2*	77.6243	46.4879	0.739653	0.01255
NM_001206402.1	*TRIM31*	12.4238	29.6212	−1.25352	0.0002
XM_003482441.2	*SCLT1*	2.56295	4.23645	−0.72505	0.0497

* The fold change represents the ratio of the F18 *E. coli* resistant group (SR) to the F18 *E. coli* sensitive group (SS).

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
