# Peer review of "Regulation and Molecular Mechanism of TLR5 on Resistance to Escherichia coli F18 in Weaned Piglets"

_animals, 2019, doi:10.3390/ani9100735_

Round 1

Reviewer 1 Report

The manuscript entitled “Regulation and molecular mechanism of TLR5 on resistance to E. coli F18 in weaned piglets” by Dai and colleagues described TLR5 plays an important regulatory role on E. coli invasion. Low expression of TLR5 inhibited the immune response and decreased cell damage, which was conducive to the resistance to E. coli infection. Overall, the majority of the experiment designs are good. Basically, this work significantly contributes to understand host-viral interaction. The manuscript is well written and ready for publication unless a minor revise.

Minor points:

All figures are blurred.

Author Response

Point 1: All figures are blurred.

Response 1: Thanks for your comment. We adjusted each image to make it clearer, hoping to meet the requirements of the manuscript.

Reviewer 2 Report

The manuscript “Regulation and molecular mechanism of TLR5 on resistance to E. coli F18 in weaned piglets” (ID: animals-575258) show interesting research issues. The introduction of the manuscript has been well developed and introduces the reader to the conditions associated with the importance of porcine TLR5 gene, both in the immune response as well as in relation to the phenomenon of resistance to bacterial infections. The goal has been clearly defined and applies molecular mechanism of TLR5 gene regulation on E. coli resistance in weaned piglets specific pig cross kept in China. The research methods are acceptable.

Specific comments

Line 21-22. The results showed that TLR5 expression level was significantly

higher in sensitive than in resistant piglets.

Line 30. Conclusionsof abstract should clearly explain the importance of discoveries (including for practice).

Line 31. Please addend in Keywords word ‘diarrhea’

Line 71. Was the animal experiment under the supervision of any Ethics Committee? Was the euthanasia of animals carried out after prior approval of the Ethics Committee?

Line 82. Country and manufacturer must be provided for the software used.

Line 101-102. Why the obtained cDNA was stored in 4oC? How long the cDNA was stored?

Line 134. Please enter the name and manufacturer of fluorescence microscope.

Line 190. Please enter the name and manufacturer of microscope.

Line 244. Was the statistical analysis one-way? What post-hoc test was used?

Line 256. Please provide explanations in the title „…piglets resistant (SR) and sensitive (SS)to E. coli F18”

Line 303. „…after bacterial stimulation”. In the context of metodic, it is quite vague.

Line 306. In what sense was the word used infection ? Quite imprecise sentence.

Line 343. I suggest moving C and D to supplementary material.

Generally, describing the data, indicate the level of significance in parentheses (please add).

Line 364. Too much generalization. In addition, which E. coli strains are concerned, at what age of piglets? Please change.

Generally the discussion is enough.

Line 408-409. The research methods on the basis of which the discoveries were achieved are unnecessary in the section “Conclusion”.

Author Response

Point 1: Line 21-22. The results showed that TLR5 expression level was significantly higher in sensitive than in resistant piglets.

Response 1: Thanks for your comment. We are so sorry that we did not described clearly and we rewrote the manuscript “The results showed that TLR5 expression level in duodenum and jejunum were significantly higher in E. coli F18-sensitive than in E. coli F18-resistant piglets”.

Point 2: Line 30. Conclusions of abstract should clearly explain the importance of discoveries (including for practice).

Response 2: Thank you for your advice. We have added the description “In conclusion, this study preliminarily revealed the molecular mechanism of TLR5 gene regulating the resistance of piglets to Escherichia coli, and provided a new candidate gene for screening Escherichia coli resistance markers in pigs”.

Point 3: Line 31. Please addend in Keywords word ‘diarrhea’

Response 3: Thank you for your advice. We have added “Diarrhea” in keywords.

Point 4: Line 71. Was the animal experiment under the supervision of any Ethics Committee? Was the euthanasia of animals carried out after prior approval of the Ethics Committee?

Response 4: Thanks for your comment. The animal experiment was under the supervision of Ethics Committee. And the euthanasia of animals was carried out after prior approval of the Ethics Committee. We added the Ethics statement in manuscript as following:

Animal experiments were approved by the Institutional Animal Care and Use Committee of the Yangzhou University Animal Experiments Ethics Committee (permit number: SYXK [Su] IACUC 2012-0029). All experimental procedures were performed in accordance with the Regulations for the Administration of Affairs Concerning Experimental Animals approved by the State Council of the People’s Republic of China. Pigs were euthanized by pentobarbital sodium injection (dose of 100 mg/kg).

Point 5: Line 82. Country and manufacturer must be provided for the software used.

Response 5: Thank you for your advice. We have added country and manufacturer in manuscript “(University of Maryland, USA)”.

Point 6: Line 101-102. Why the obtained cDNA was stored in 4°C? How long the cDNA was stored?

Response 6: Thanks for your question. In fact, we stored cDNA in 4°C no more than 1 hour because we conducted the qPCR at once. We stored cDNA at −70°C for long-term preservation. To avoid ambiguity, we modified the manuscript “The resulting cDNA was stored at −70°C”.

Point 7: Line 134. Please enter the name and manufacturer of fluorescence microscope.

Response 7: Thank you for your advice. We have added name and manufacturer in manuscript “(Olympus Fluorescence Inverted Microscope IX73, Olympus Corporation, Japan)”.

Point 8: Line 190. Please enter the name and manufacturer of microscope.

Response 8: Thank you for your advice. We have added name and manufacturer in manuscript “(Operetta CLS, PerkinElmer, USA)”.

Point 9: Line 244. Was the statistical analysis one-way? What post-hoc test was used?

Response 9: Yes, differences in gene expression and cytokine levels were analyzed by One-Way ANOVA. And LSD was used for post-hoc test. We added it in revised manuscript.

Point 10: Line 256. Please provide explanations in the title „…piglets resistant (SR) and sensitive (SS) to E. coli F18”

Response 10: Thank you for your advice. We have provided the explanations in 2.1. “We screened three resistant and three sensitive animals of similar birth weight, weaning weight, body shape, and coat color using a V-type secretion system [22].” Therefore, the explanations were not provided in the table title.

Point 11: Line 303. „…after bacterial stimulation”. In the context of metodic, it is quite vague.

Response 11: Thanks for your comment. The method and dosage of bacterial stimulation were introduced in detail in the Materials and Methods section 2.8. “Each treatment sample had three repeats. E. coli F18ab, E. coli F18ac, and E. coli K88ac fimbriae standard strains were inoculated to LB culture medium and incubated for 12 h at 200 r/min on a rocking platform. The bacteria were diluted to 1.0 × 109 CFU/mL with cell culture medium. Culture medium (1.0 mL) was added to 12-well culture plates and incubated for 3 h”. To make it clear, we modified the manuscript with “and the levels of cytokines in the interference cells after stimulated by E. coli F18ab, E. coli F18ac, and E. coli K88ac tended to the levels of control cells without bacterial stimulation”.

Point 12: Line 306. In what sense was the word used infection ? Quite imprecise sentence.

Response 12: Thanks for your comment. We replaced the word “infection” with “stimulation” to make the sentence more precise.

Point 13: Line 343. I suggest moving C and D to supplementary material.

Response 13: Thank you for your suggestion. In Figure 6, we predicted three core promoter regions of TLR5, (C) showed the prediction of the core promoter region based on BDGP software. (D) showed the analysis of acting elements in the promoter region. Both of them were highly related to (A) and (B), so we hope that they could be retained in Figure 6.

Point 14: Generally, describing the data, indicate the level of significance in parentheses (please add).

Response 14: Thank you for your suggestion. We have added significance in parentheses.

Point 15: Line 364. Too much generalization. In addition, which E. coli strains are concerned, at what age of piglets? Please change.

Response 15: Thank you for your suggestion. We have chenged the description in revised manuscript “Weaned piglets (5 to 6 weeks of age) are not sensitive to hemolytic E. coli, because the receptor is not expressed and antibodies in breast milk confer protection [29]”.

Point 16: Line 408-409. The research methods on the basis of which the discoveries were achieved are unnecessary in the section “Conclusion”.

Response 16: Thank you for your suggestion. We have modified the manuscript to make it more refined.

Reviewer 3 Report

The manuscript described how TLR5 regulate resistance to E. coli F18 infection in weaned piglets. At the moment, the experiment design were confusing and the “Materials and Methods and Results” were not clearly written. In addition. cell localization of TLR5 (immunological regulators) were conducted in standard cell line not from piglets, how the knowledge gained from standard cell line in vitro reflects what happen in intestinal tract of piglets, which . Because piglets were already included in the study, why not just challenged with E. coli and carried localization experiment with tissue samples. 

Additional comments

Title: full name of E. coli please. Based on the cell-line experimental content stated above, not quite convinced it can be called  in weaned piglets. 

L61, you only studied intestinal tissues not other organs, so don’t think this can be called systematically.

Material and Methods are lengthy 4 pages? Can any standard methods go supplemental materials?

L66-71,

be consistent in writing pig vs pig lets; need to describe age and weight of weaned piglets at the time of experiment. Also, how about gender of piglets? Would this affect results? All pigs were housed in the same environment? Please be specific. E.g. Were pigs housed individually or F18-resistant pigs housed together, but separated from F18-sensitive pigs? What diet were they fed? Were piglets experimentally challenged with E. coli ? References for “at the age of 35days, when piglets are most susceptible to E. coli F18”. Need Ethical approval.

L73, qPCR. What is BSP standard for?

L76 why was transcriptome analysis only done in duodenum? How about jejunum?

L78, value for good RNA quality?

L77-86, was RNA extraction method standard or developed by the authors? If the former, please indicate references and shorten the description.

Section 2.3 and 2.4 are confusing, describing same thing?

Section 2.5 what was RT-PCR done for?

L113, total protein were extracted from where?

L136, what were IPEC-J2 cells? OK, saw L150 description, please move the description to L136

L245, what was treatment referred here?

Results, I’d suggest to add 1-2 sentences at the beginning of each paragraph to state rational of doing the experiment. In this case, the writing can have better flow. Currently, I’m afraid that I lost in lots of data and don’t quite understand what these data were generated for what reason. In my opinion, this results section was quite short (6 pages) which include 1 table and 7 figures.

Author Response

Point 1: Title: full name of E. coli please. Based on the cell-line experimental content stated above, not quite convinced it can be called in weaned piglets. 

Response 1: Thank you for your advice. We have added the full name of “E. coli” in title. In this study, the TLR5 gene was screened out by RNA-seq of duodenum derived from E. coli F18-resistant and -sensitive Sutai weaned piglets, which indicated the potential role of the TLR5 gene in regulating E. coli F18 infection. To further validate and investigate the functional role of TLR5 expression and molecular mechanisms involved in regulating E. coli F18 infection, the experiments were thus performed in the IPEC-J2 cells that are widely used in gene function validation. All the experiments of this study was aimed to prove that TLR5 gene indeed played an important role in regulating E. coli stimulation. Therefore, “in weaned piglets” was used in the title.

Point 2: L61, you only studied intestinal tissues not other organs, so don’t think this can be called systematically.

Response 2: Thanks for your comment. Resistance to E. coli F18 depends on expression of receptors on intestinal epithelial cells, and individual immunity. Recent study reported the mechanisms that regulate the interaction between the immune system and the microbiota, focusing on the role of resident intestinal bacteria in the development of immune responses (Kamada and Núñez, 2014). Toll-like receptor (TLR) family recognizes conserved microbial structures, such as bacterial lipopolysaccharide and viral double-stranded RNA, and activates signaling pathways that result in immune responses against microbial infections (Barton and Medzhitov, 2003). On this basis, our study focused on the relationship of TLR5 expression and intestinal immune response, and we have deleted the “systematically”.

Kamada, N. & Núñez, G. Regulation of the immune system by the resident intestinal bacteria. Gastroenterology 146, 1477–1488 (2014).

Barton, G. M. & Medzhitov, R. Toll-like receptor signaling pathways. Science 300, 1524–1525 (2003).

Point 3: Material and Methods are lengthy 4 pages? Can any standard methods go supplemental materials?

Response 3: Thanks for your comment. We aimed to describe each experiment as clear as possible, now we have modified the manuscript to make them more concise.

Point 4: L66-71,be consistent in writing pig vs pig lets; need to describe age and weight of weaned piglets at the time of experiment. Also, how about gender of piglets? Would this affect results? All pigs were housed in the same environment? Please be specific. E.g. Were pigs housed individually or F18-resistant pigs housed together, but separated from F18-sensitive pigs? What diet were they fed? Were piglets experimentally challenged with E. coli ? References for “at the age of 35days, when piglets are most susceptible to E. coli F18”. Need Ethical approval.

Response 4: Thanks for your questions. The identification of E. coli F18-resistant and -sensitive Sutai pigs was the result our previous research (Bao et al., 2012; Ye et al., 2012), and all factors were taken into account. We have provided relevant references. In addition, we have added the Ethical statement in manuscript.

Bao W B, Ye L, Pan Z Y, et al. Microarray analysis of differential gene expression in sensitive and resistant pig to Escherichia coli F18[J]. Animal genetics, 2012, 43(5): 525-534.

Ye L, Su X, Wu Z, et al. Analysis of differential miRNA expression in the duodenum of Escherichia coli F18-sensitive and-resistant weaned piglets[J]. PLoS One, 2012, 7(8): e43741.

Point 5: L73, qPCR. What is BSP standard for?

Response 5: Thanks for your question. We modified the description in revised manuscript “Quantitative PCR (qPCR) primers for TLR5 based on its coding sequence, PCR primers for the promoter region of TLR5 and Bisulfite Sequencing PCR (BSP) primers for methylation detection were designed”. 

Point 6: L76 why was transcriptome analysis only done in duodenum? How about jejunum?

Response 6: Thanks for your question. In this study, we only conducted the transcriptome analysis of duodenum because our previous research revealed the differential miRNA expression in the duodenum of Escherichia coli F18-sensitive and -resistant weaned piglets (Ye et al, 2012), which used duodenum as sample. Of course, jejunum is also an important tissue for study on intestinal immunity. We will consider using jejunum as a sample for high throughput analysis in future studies.

Ye L, Su X, Wu Z, et al. Analysis of differential miRNA expression in the duodenum of Escherichia coli F18-sensitive and-resistant weaned piglets[J]. PLoS One, 2012, 7(8): e43741.

Point 7: L78, value for good RNA quality?

Response 7: Thanks for your question. We have strictly controlled the quality of RNA samples. The quality control standards mainly include the following three aspects. Agarose gel electrophoresis: sample RNA integrity and DNA contamination were analyzed. NanoPhotometer spectrophotometer: Detection of RNA purity (OD260/280 and OD260/230 ratio). Agilent 2100 bioanalyzer: Accurate detection of RNA integrity. In conclusion, good RNA quality was ensured before sequencing. We did not add RNA quality control standards in manuscript to make it more concise.

Point 8: L77-86, was RNA extraction method standard or developed by the authors? If the former, please indicate references and shorten the description.

Response 8: Thanks for your comment. This was the standard process for building RNA libraries. We have indicated the reference and shorten the description according to your advice.

Point 9: Section 2.3 and 2.4 are confusing, describing same thing?

Response 9: Thanks for your question. Section 2.3 (now is Section 2.4) described the construction of RNA library for sequencing, in which RNA was extracted from the duodenum of resistant individuals and three sensitive individuals. Section 2.4 (now is Section 2.5) described the process of gene expression detection, in which RNA was extracted from duodenum and jejunum of piglets and IPEC-J2 cells. We have modified the manuscript to make them more clear.

Point 10: Section 2.5 what was RT-PCR done for?

Response 10: Thanks for your question. It was conducted for detecting the gene expression in duodenum and jejunum of piglets and IPEC-J2 cells. We added the description in our revised manuscript.

Point 11: L113, total protein were extracted from where?

Response 11: Thanks for your question. It was conducted for detecting the protein expression in duodenum and jejunum of piglets and IPEC-J2 cells. We added the description in our revised manuscript. 

Point 12: L136, what were IPEC-J2 cells? OK, saw L150 description, please move the description to L136

Response 12: Thank you for your advice. We have moved the description according to your suggestion.

Point 13: L245, what was treatment referred here?

Response 13: Thanks for your question. We meant that all experimental sample had three replicates. To avoid ambiguity, we modified the manuscript “All experimental sample had three replicates, and the results were presented as mean ± standard deviation”.

Point 14: Results, I’d suggest to add 1-2 sentences at the beginning of each paragraph to state rational of doing the experiment. In this case, the writing can have better flow. Currently, I’m afraid that I lost in lots of data and don’t quite understand what these data were generated for what reason. In my opinion, this results section was quite short (6 pages) which include 1 table and 7 figures.

Response 14: Thank you for your suggestions. We have modified the results according to your advice.